# Melioidosis Queensland: An analysis of clinical outcomes and genomic factors

Ian Gassiep[1,2,3]*, Delaney Burnard[4], Budi Permana[1,5], Michelle J. Bauer[1], Thom Cuddihy[1], Brian M. Forde[1], Mark D. Chatfield[1], Weiping Ling[1], Robert Norton[6,7], Patrick N. A. Harris[1,3]

1 The University of Queensland, Faculty of Medicine, UQ Centre for Clinical Research, Herston, Queensland, Australia, 2 Department of Infectious Diseases, Mater Hospital Brisbane, South Brisbane, Queensland, Australia, 3 Pathology Queensland, Royal Brisbane & Women's Hospital, Herston, Queensland, Australia, 4 Queensland Cyber Infrastructure Foundation, Brisbane, Queensland, Australia, 5 Herston Infectious Diseases Institute, Metro North Health, Queensland, Australia, 6 Pathology Queensland, Townsville University Hospital, Townsville, Queensland, Australia, 7 Faculty of Medicine, University of Queensland, Brisbane, Queensland, Australia

* i.gassiep@uq.edu.au

**Data Availability Statement:** Data generated from this study is available under the NCBI accession PRJNA960936.

## Abstract

### Background

The clinical and genomic epidemiology of melioidosis varies across regions.

### Aim

To describe the clinical and genetic diversity of *B. pseudomallei* across Queensland, Australia.

### Methods

Whole genome sequencing of clinical isolates stored at the melioidosis reference lab from 1996–2020 was performed and analysed in conjunction with available clinical data.

### Results

Isolates from 292 patients were analysed. Bacteraemia was present in 71% and pneumonia in 65%. The case-fatality rate was 25%. Novel sequence types (ST) accounted for 51% of all isolates. No association was identified between the variable virulence factors assessed and patient outcome. Over time, the proportion of First Nation's patients declined from 59% to 26%, and the proportion of patients aged >70 years rose from 13% to 38%.

### Conclusion

This study describes a genomically diverse and comparatively distinct collection of *B. pseudomallei* clinical isolates from across Queensland, Australia. An increasing incidence of melioidosis in elderly patients may be an important factor in the persistently high case-fatality in this region and warrants further investigation and directed intervention.

**Funding:** IG received funding for the whole genome sequencing provided by a Royal Australasian College of Physicians Queensland Regional Committee Research Development Grant and the Pathology Queensland Study and Education Committee (SERC 6145). The funders had no role in study design, data collection and analysis, decision to publish, or preparation of the manuscript.

**Competing interests:** The authors have declared that no competing interests exist.

## Author summary

*Burkholderia pseudomallei* is an environmental bacteria found in the soil and water of tropical and subtropical regions. This organism causes melioidosis, an infectious disease that often results in pneumonia and bloodstream infections. Previous studies have suggested a relationship between specific bacterial genetic factors and disease outcomes. This study reports on the clinical and bacterial genetic factors associated with disease outcomes in Queensland, Australia. A total of 292 patients and the associated bacterial isolates obtained from them during their hospital admission were analysed. The majority of patients presented with pneumonia and had a bloodstream infection. The case-fatality rate improved slightly over time but remains higher than other regions in Australia. Over the period of the study the proportion of First Nation's people affected decreased significantly. Notably, for reasons yet to be determined, the proportion of patients aged over 70 years increased over time. The bacterial genetic data demonstrate a high level of diversity across the region, however there was no association between previously described virulence genes and clinical outcomes in our cohort. This study creates a platform for further state-wide melioidosis research in Queensland.

## Introduction

*Burkholderia pseudomallei* is an environmental organism found in the soil and water of many tropical and subtropical regions [1]. The organism is a Gram negative bacterium and the cause of melioidosis. This is an infectious disease which is commonly acquired through inhalation or inoculation, often presents with bacteraemia, and has a case-fatality rate which varies from 6−42% [2,3].

The clinical epidemiological data of melioidosis in relation to risk factors for disease and mortality in high burden regions such as Thailand and Northern Australia is robust [3,4]. However, although multiple regions have reported on the genomic epidemiology in relation to disease manifestation and patient outcomes, there remains a substantial knowledge gap. Genomic data from each region continue to demonstrate substantial organism diversity. Most recent publications, which have identified the multilocus sequence types (MLST) of their *B. pseudomallei* isolate collections, have reported high rates of novel or previously not described sequence types [5–9]. Given the current knowledge of the highly recombinant nature of the organism this is not surprising.

In terms of genetic association with disease and patient outcomes, only a limited number of virulence factors have been assessed. These include the filamentous hemagglutinin protein (fhaB3), Yersinia-like fimbriae (YLF) and *Burkholderia thailandensis*-like flagellum and chemotaxis (BTFC) gene clusters, and the *Burkholderia* intracellular motility factor A ($bimA_{Bm}$) allele variant [10,11]. Additionally, a study assessing the correlation between disease severity and outcomes with respect to the lipopolysaccharide (LPS) genotypes of *B. pseudomallei* has been performed [12].

Notably, there are limited published data on the association between *B. pseudomallei* virulence factors and disease. In one study the filamentous hemagglutinin gene *fhaB*3 was found to be associated with bacteraemia [10]. Furthermore, this study found no statistically significant association between the YLF/BTFC gene clusters and clinical features [10]. The $bimA_{Bm}$ variant has been shown to be associated with neurological disease in Australia and mortality in

India [10,11,13]. Finally, all LPS genotypes have previously demonstrated no association with bacteraemia, septic shock, nor mortality [12].

Queensland is the North Eastern state of Australia. It is the second largest state by area. Climate conditions range from the temperate south, to arid west, and the tropical north [14]. As of 2021 the total population was approximately 5.2 million residents. First Nations people accounted for 4.6% of the population. The median age was 38 years with a male to female ratio of 0.97. Employment in Agriculture, Forestry, and Fishing accounted for 2.7% of the population, with men representing two-thirds of these employees [15]. The primary land use industries in Queensland were diverse including but not limited to cattle, sugar cane, and sorghum [16].

The Pathology Queensland Melioidosis Reference Laboratory has prospectively collected clinical *B. pseudomallei* isolates from most regions of Queensland from 1996 to present day. Therefore, this study aimed to provide clinical and genomic epidemiological data for this large and diverse geographical region. Furthermore, we aim to assess the association between the aforementioned virulence factors and clinical outcomes.

## Methods

### Ethical statement

This study received ethical approval from the Royal Brisbane & Women's Hospital Ethics Committee (LNR/2020/QRBW/65573), with site-specific authority obtained from the Townsville Hospital and Health Service and approval under the Queensland Public Health Act.

The Pathology Queensland laboratory based at Townsville University Hospital in North Queensland is the state melioidosis reference laboratory. Isolates from all regions of Queensland, with the exception of Cairns and Hinterland & Torres and Cape, have routinely been referred to and stored at the reference laboratory since 1996.

All patients with culture-confirmed melioidosis identified between 1 January 1996 and 31 December 2020 were included. Where possible, retrospective clinical details were obtained from the hospital medical records. Death attributable to melioidosis within 120 days was included. Data from the Australian Bureau of Statistics and Queensland Health were used to calculate the incidence [15,17]. Incidences were visualised using geojsonio v0.11.1 R package (https://github.com/ropensci/geojsonio) and ArcGIS Pro version 3.1, over the map of hospital and health service area boundaries across Queensland downloaded from Queensland Spatial Catalogue, Department of Resources, State of Queensland [18]. The data included in this study include those from the Townsville Hospital melioidosis cohort which is described elsewhere (manuscript in preparation).

The bacterial isolates from all 292 patients included in this study were recovered from -80°C storage and subcultured onto horse blood agar (HBA). DNA extraction was performed with the QIAGEN DNAeasy ultra-pure DNA extraction kit according to manufacturer's instructions. Sequencing libraries were generated using the Nextera Flex DNA library preparation kit and sequenced on the MiniSeq System (Illumina Inc., San Diego, CA, USA) on a high output 300 cycle cartridge according to the manufacturer's instructions. Data generated from this study is available under the NCBI accession PRJNA960936.

Raw Illumina reads were trimmed with Trimmomatic v0.36 [19]. Read quality was assessed with multiQC v1.11 and genomes were assembled with SPAdes v3.14.0 [20,21]. Reads were mapped with BWA-MEM v0.7.17 [22]. Sequence types (STs) were determined with multi-locus sequence typing (MLST) (https://github.com/tseemann/mlst) and reads mapped to alleles retrieved from pubMLST (access date: 2023-04-25) [23]. Genotypic antimicrobial resistance was determined with ArDaP v1.8.2 [24].

A custom database was created and run with ABRicate (https://github.com/tseemann/abricate) to screen for virulence determinants including the LPS profile, *fhaB3* gene, YLF/BTFC gene clusters, and detection of the *bimA*<sub>Bm</sub> allele variant, S1 Table [10,12].

A total of 174 previously published genomes were downloaded and used to provide global context of study isolates, S2 Table. The global and local maximum likelihood trees were constructed based on single nucleotide polymorphism (SNP) variants extracted from the genomic reads compared against K96243 reference genome (accessions: SAMEA1705938) using genome Analysis Toolkit (GATK v4.2) and FastTree v2.1.10 wrapped in SPANDx v4.03 under the default settings. Phylogenetic trees were visualised using and ggtree v2.4.1 R package [25,26].

Isolates with an E-test (bioMérieux, Marcy-l'Etoile, France) minimum inhibitory concentration (MIC) result were included in the analysis. Results were interpreted using the European Committee on Antimicrobial Susceptibility Testing (EUCAST) criteria.

### Statistical analysis

Data were analysed using Stata version 16 (StataCorp, College Station, TX, USA). Categorical variables were analysed using chi-squared or Fisher's exact test. The Cochrane-Armitage test was used to analyse trends over time. A threshold of $p \le 0.1$ in a simple logistic regression model was used for inclusion of covariate in the multivariate logistic regression model.

### Results

There were 292 clinical isolates sequenced and matched to demographic and clinical data. The majority of infections (67%) occurred in men, Table 1. First Nations (FN) patients accounted for 30% of the cohort and were substantially younger than their non-FN counterparts, 45 vs 61 years old, respectively (Table 2). Eleven cases occurred in paediatric patients (age <18). Limited clinical data were available, however 71% (197/279) of patients were bacteraemic and 65% (185/283) presented with pneumonia. The overall case-fatality rate was 25% (69/280).

Bivariate analysis demonstrated an association with mortality for both bacteraemia and pneumonia. Bacteraemic patients had a 31% case-fatality rate compared with 8% in non-bacteraemic patients, p: <0.001. Similarly, 29% of patients with pneumonia died compared with 15% of patients without pneumonia, p: 0.01. However, the association with pneumonia and case-fatality was not significant on multivariate analysis (S3A and S3B Table). Mortality was 29% in 1996–2004 and 21% in 2013–2020, p: 0.43.

Melioidosis cases occurred across 6 main health services including Townsville, Mornington Island, Mount Isa, Mackay, Ingham, and Bowen, (S1 Fig and S4 Table). The greatest proportion of cases occurred in Townsville (56%), Table 3, and Fig 1. Both Mornington Island and Mount Isa had the greatest proportion of FN patients and were the youngest cohorts. Bowen reported the highest rate of bacteraemia 100% (14/14), however only 21% (3/14) died. In contrast, Mornington Island reported 66% (21/33) bacteraemic and 30% (9/33) dead (S5 Table).

With regards to sequence type, 113 previously undescribed STs were identified. These novel STs accounted for 51% of all isolates. The most prevalent known STs were ST 252 with 12% (35/292), ST 283 with 7% (21/292), and all others identified in less than 5% of isolates. The most prevalent novel ST, labelled TSV (Townsville), was ST TSV13 at 3% (8/292), with all others occurring in less than 2% of isolates. Over the course of the study period the prevalence of novel STs increased from 28% to 48% of isolates (p: <0.001). There was no association between ST and case-fatality.

Fig 2 demonstrates the genomic diversity of the isolates included in this study in relation to international isolates (panel A) and within Queensland (panel B). It is possible that 3 cases

**Table 1. Factors associated with mortality.**

| | Total | Alive | Dead | p-value |
|---|---|---|---|---|
| | N = 292 | N = 211 | N = 69 | |
| **Age, median (IQR)** | 57.5 (46–68) | 57 (45–67) | 60 (47–75) | 0.1 |
| **Age groups** | | | | |
| 18–49 | 93 (32%) | 72 (79%) | 19 (21%) | 0.1 |
| 50–69 | 130 (45%) | 95 (78%) | 27 (22%) | |
| ≥70 | 69 (24%) | 44 (66%) | 23 (34%) | |
| **Age >50** | 199 (68%) | 139 (74%) | 50 (26%) | 0.3 |
| **First Nation** | | | | |
| Yes | 87 (30%) | 65 (77%) | 19 (23%) | 0.6 |
| No | 205 (70%) | 146 (75%) | 50 (25%) | |
| **Sex** | | | | |
| Female | 97 (33%) | 71 (75%) | 24 (25%) | 0.9 |
| Male | 195 (67%) | 140 (76%) | 45 (24%) | |
| **Region** | | | | |
| Townsville | 162 (63%) | 119 (77%) | 36 (23%) | 0.9 |
| Mornington Island | 33 (13%) | 21 (70%) | 9 (30%) | |
| Mount Isa | 17 (7%) | 14 (82%) | 3 (18%) | |
| Mackay | 16 (6%) | 13 (81%) | 3 (19%) | |
| Ingham | 15 (6%) | 11 (79%) | 3 (21%) | |
| Bowen | 14 (5%) | 11 (79%) | 3 (21%) | |
| **Diagnosis year** | | | | |
| 1996–2004 | 116 (40%) | 77 (71%) | 31 (29%) | 0.4 |
| 2005–2012 | 68 (23%) | 49 (77%) | 15 (23%) | |
| 2013–2020 | 108 (37%) | 85 (79%) | 23 (21%) | |
| **Bacteraemia** | | | | |
| Yes | 197 (71%) | 132 (68%) | 62 (32%) | <0.001 |
| No | 82 (29%) | 69 (92%) | 6 (8%) | |
| **Pneumonia** | | | | |
| Yes | 185 (65%) | 128 (71%) | 52 (29%) | 0.01 |
| No | 98 (35%) | 79 (85%) | 14 (15%) | |
| **Novel ST** | | | | |
| Yes | 150 (51%) | 116 (80%) | 30 (20%) | 0.1 |
| No | 142 (49%) | 95 (71%) | 39 (29%) | |
| **LPSA** | | | | |
| Yes | 226 (77%) | 164 (75%) | 54 (25%) | 0.9 |
| No | 66 (23%) | 47 (76%) | 15 (24%) | |
| *fhaB3* | | | | |
| Yes | 237 (81%) | 173 (75%) | 58 (25%) | 0.7 |
| No | 55 (19%) | 38 (78%) | 11 (22%) | |
| **YLF** | | | | |
| Yes | 156 (53%) | 107 (72%) | 42 (28%) | 0.1 |
| No | 136 (47%) | 104 (79%) | 27 (21%) | |
| **BTFC** | | | | |
| Yes | 132 (45%) | 100 (79%) | 27 (21%) | 0.2 |
| No | 160 (55%) | 111 (73%) | 42 (27%) | |
| $bimA_{Bm}$ | | | | |
| Yes | 54 (18%) | 42 (81%) | 10 (19%) | 0.3 |
| No | 238 (82%) | 169 (74%) | 59 (26%) | |

**Table 2. First Nation status and association with demographic, clinical, and virulence factors.**

|  | First Nation | Non-First Nation | *p-value* |
|---|---|---|---|
|  | N = 87 | N = 205 |  |
| **Age, median (IQR)** | 47 (38–54) | 63 (53–73) | *<0.001* |
| **Age groups** |  |  |  |
| 18–49 | 51 (59%) | 42 (21%) | *<0.001* |
| 50–69 | 32 (37%) | 98 (48%) |  |
| ≥70 | 4 (5%) | 65 (32%) |  |
| **Age ≥50** | 35 (40%) | 164 (80%) | *<0.001* |
| **Male** | 52 (60%) | 143 (70%) | *0.1* |
| **Region** |  |  |  |
| Mornington Island | 32 (41%) | 1 (1%) | *<0.001* |
| Townsville | 31 (40%) | 131 (73%) |  |
| Mount Isa | 11 (14%) | 6 (3%) |  |
| Ingham | 3 (4%) | 12 (7%) |  |
| Mackay | 1 (1%) | 15 (8%) |  |
| Bowen | 0 (0%) | 14 (8%) |  |
| **Diagnosis year** |  |  |  |
| 1996–2004 | 51 (59%) | 65 (32%) | *<0.001* |
| 2005–2012 | 13 (15%) | 55 (27%) |  |
| 2013–2020 | 23 (26%) | 85 (42%) |  |
| **Bacteraemia** | 61 (72%) | 136 (70%) | *0.8* |
| **Pneumonia** | 58 (72%) | 127 (63%) | *0.2* |
| **Novel-ST** | 45 (52%) | 105 (51%) | *0.9* |
| **LPSA** | 69 (79%) | 157 (77%) | *0.6* |
| ***fhaB3*** | 68 (78%) | 169 (82%) | *0.4* |
| **YLF** | 23 (26%) | 133 (65%) | *<0.001* |
| **BTFC** | 61 (70%) | 71 (35%) | *<0.001* |
| ***bimA*$_{Bm}$** | 18 (21%) | 36 (18%) | *0.5* |

were imported from Asia. These isolates cluster with those from the Philippines (4455 SNPs). A geospatial summary of the most common sequence types in relation to region of Queensland is also represented. The 6 most common STs appear to be associated with the Townsville region, and all ST 64 isolates were found in Mount Isa.

Virulence factor analysis identified the YLF gene cluster in 53%, BTFC in 45%, and 1% of isolates were identified as not carrying either gene cluster. The *fhaB3* gene was found in 81%, and *bimA*$_{Bm}$ in 18%. Lipopolysaccharide typing identified LPS A in 77%, LPS B2 in 1%, LPS B1 0%, and 21% were non-typeable.

Patients with the LPS A genotype were less likely to be bacteraemic (RR: 0.4, 95% CI: 0.2– 0.8, p: 0.04) (S6 Table). Similarly, the risk of pneumonia was lower in patients with isolates with *fhaB3* (RR: 0.78, 95% CI: 0.66–0.92, p: 0.01) (S7 Table).

The BTFC gene cluster was more likely to be found in novel STs (64%) compared to the YLF gene cluster (40%), p: <0.001 (S8 Table). BTFC was also more likely to be found in isolates from FN patients, Table 2. Of the isolates with the *bimA*$_{Bm}$ gene, 83% were found in novel STs, p: <0.001 (S9 Table). No clinical association was found with the *bimA*$_{Bm}$ gene. Notably, an assessment of central nervous system infections could not be performed due to limited clinical data.

Phenotypic antimicrobial susceptibility results were available for the following agents, ceftazidime (CAZ), meropenem, trimethoprim-sulfamethoxazole (SXT), and doxycycline (DOX).

**Table 3. Comparison of regions with more than 10 cases over the study period.**

| | BOW | ING | MK | MI | MOR | TSV | *p-value* |
|---|---|---|---|---|---|---|---|
| **Clinical** | | | | | | | |
| Number of cases | 14 (5) | 15 (5) | 16 (5) | 17 (6) | 33 (11) | 162 (56) | - |
| Male | 11 (79) | 11 (73) | 14 (82) | 14 (82) | 20 (61) | 110 (68) | *0.7* |
| First Nation | 0 | 3 (20) | 1 (6) | 11 (65) | 32 (97) | 31 (19) | *<0.001* |
| Age>50 | 12 (86) | 13 (87) | 14 (88) | 7 (41) | 17 (52) | 113 (70) | *0.004* |
| Pneumonia | 8 (62) | 11 (73) | 10 (63) | 10 (59) | 17 (61) | 105 (66) | *0.9* |
| Bacteraemia | 14 (100) | 9 (64) | 12 (86) | 10 (63) | 21 (66) | 106 (69) | *0.7* |
| Dead | 3 (21) | 3 (21) | 3 (19) | 3 (18) | 9 (30) | 36 (23) | *0.9* |
| **Genomic** | | | | | | | |
| Novel ST | 13 (93) | 9 (60) | 10 (63) | 14 (82) | 19 (58) | 64 (40) | *<0.001* |
| YLF | 7 (50) | 7 (47) | 11 (69) | 3 (18) | 3 (9) | 108 (67) | *<0.001* |
| BTFC | 7 (50) | 8 (53) | 5 (31) | 14 (82) | 28 (85) | 53 (33) | *<0.001* |
| *fhaB3* | 8 (57) | 12 (80) | 12 (75) | 16 (94) | 31 (94) | 129 (80) | *0.04* |
| LPS A | 10 (71) | 11 (73) | 13 (81) | 17 (100) | 31 (94) | 117 (72) | *0.008* |
| $bimA_{Bm}$ | 7 (50) | 3 (20) | 1 (6) | 3 (17) | 7 (21) | 25 (15) | *0.05* |

Locations: BOW: Bowen; ING: Ingham; MK: Mackay; MI: Mount Isa; MOR: Mornington Island; TSV: Townsville

N and row percentages are presented in each cell

Only 195 isolates were tested for both CAZ and MEM. The median CAZ MIC = 1.5 mg/L (IQR: 1.0–2.0), and median MEM MIC = 1 mg/L (IQR: 0.75–1.0). No isolates were MEM resistant, and only 1 isolate was CAZ resistant (MIC = 256). SXT susceptibility was performed on 226 isolates, with a median MIC = 0.75 mg/L (IQR: 0.38–1), and only 1 resistant isolate (MIC = 12). DOX was performed on 196 isolates with a median MIC = 1 mg/L (IQR: 0.75–1.5). Nineteen isolates were resistant (MIC>2) to DOX.

Genomic antimicrobial susceptibility analysis revealed 15 isolates with a genetic determinant for potential phenotypic resistance. Twelve isolates contained a penA Ser78Phe gene mutation. However, no associated penA -78G>A promoter mutation, required for amoxicillin-clavulanic acid resistance, was detected. One isolate was doxycycline resistant by both phenotype (MIC = 6 mg/L) and genotype ($BPSL3085_{A88fs}$). This isolate was also found to have an efflux pump loss-of-function variant associated with gentamicin susceptibility ($BPSL1803$ $AmrB_{A254fs}$). One isolate demonstrated gene loss/truncation associated with an elevated trimethoprim-sulfamethoxazole (SXT) MIC ($Ptr1_{R21fs}$), however was phenotypically susceptible (MIC = 2 mg/L). The final isolate with genotypic resistance determinants was reported as resistant to ceftazidime (CAZ), meropenem (MEM), and imipenem (mecA_BX571856). Notably, the phenotypic MICs were CAZ = 1mg/L and MEM = 0.75 mg/L.

Over the study period the proportion of FN patients decreased from 59% down to 26%, p: <0.001 (Table 2). Additionally, when comparing the age of diagnosis over time, there was a significant increase in older patients diagnosed with melioidosis. From 1996–2004 only 13% (15/116) of patients were ≥70 years old. This increased to 19% (13/68) in 2005–2012, then 38% (41/108) in the period 2013–2020, p: <0.001.

## Discussion

The authors aimed to report on the clinical and genomic data from a collection of clinical *Burkholderia pseudomallei* isolates obtained from across Queensland, Australia. The isolates in this collection span a massive geographic area, with the distance between some regions

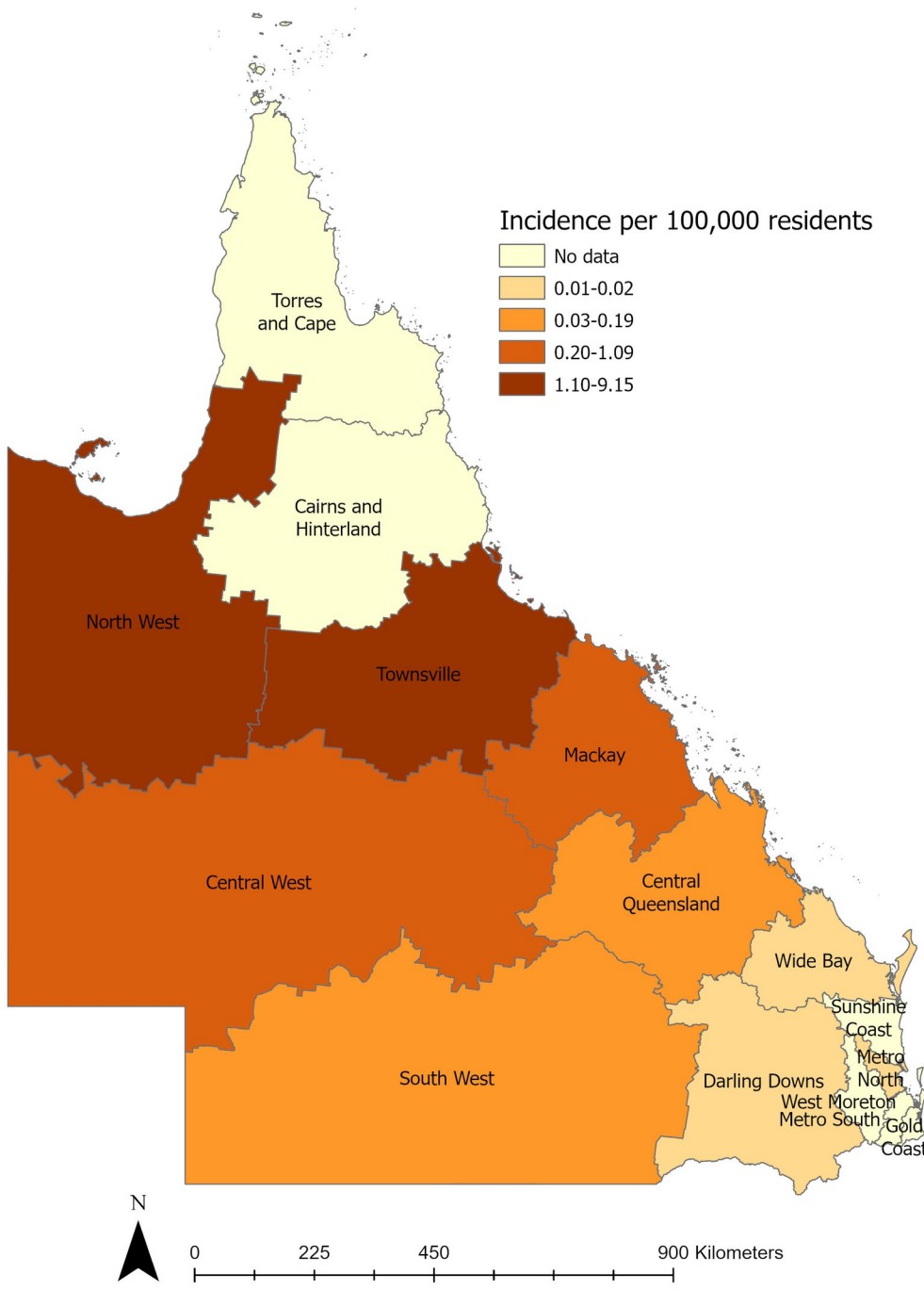

**Fig 1. Incidence of melioidosis in Queensland, Australia\*.** Torres and Cape & Cairns and Hinterland were not included in the analysis (Map created using ArcGIS Pro version 3.1 & data from Queensland Spatial Catalogue https://qldspatial.information.qld.gov.au/catalogue/custom/detail.page?fid={A4661F6D-0013-46EE-A446-A45F01A64D46}).

measuring over 1,000 kilometres. Furthermore, these isolates were collected over a 25-year period and therefore provide valuable trend over time information.

Unsurprisingly, Townsville, the most populous city included in this analysis, had the highest proportion of cases. Although, it is notable that the North West region which predominantly includes Mornington Island and Mount Isa, had a similarly high incidence per 100,000

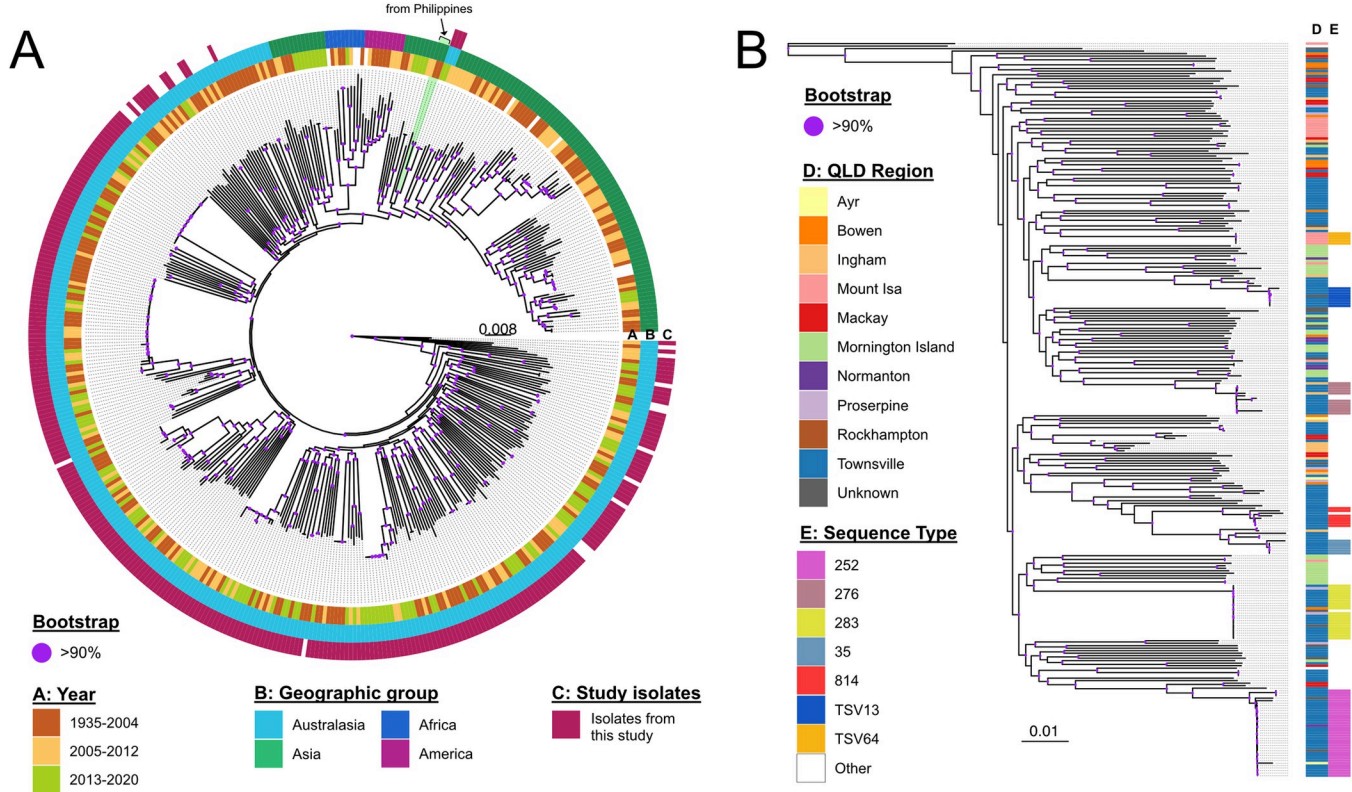

**Fig 2. Maximum likelihood phylogeny of 292 clinical *Burkholderia pseudomallei* isolates from Queensland with a global set of genomes (n = 174) based on the alignment of 176,592 SNPs.** A. B. Maximum likelihood phylogeny of the Queensland isolates based on the alignment of 166,688 SNPs. Both trees were rooted to the most ancestral *B. pseudomallei* strain (MSHR0668). Scales represent nucleotide substitutions per site.

population. There was a stark contrast between the different regions and the proportion of First Nation patients. Almost all patients from Mornington Island identified as FN compared with no FN patients in Bowen and only 1 in Mackay. Generally, this coincides with the specific demographics of each region, with 80% of Mornington Island residents identifying as FN [15]. However, it is unclear why Bowen with an FN population similar to both Ingham and Townsville had no cases in that population. Furthermore, it is unclear why there appears to be a substantial difference in patients presenting with bacteraemia in Bowen compared to all other regions. Finally, it is notable that there was a greater than 50% reduction in patients identifying as First Nation Australian over the course of the study period. These differences may be due to the relatively low number of cases across the various regions. It is also important to note that clinical risk factor data such as prevalence of diabetes mellitus or immunosuppression are missing; further limiting comparison of these regions.

The overall case-fatality rate in this study is higher than that reported from other regions of Australia [3,27]. The rate over time appears to be slower to improve in comparison to those studies. It is important to note that the proportion of older patients in our cohort has increased over the duration of the study period. Although not statistically significant, there was a 12% higher case-fatality rate in those patients over 70 years of age compared to the younger cohorts. It is unclear as to why there is an increase in melioidosis incidence in this age group, and additional longitudinal data is warranted to further assess this issue.

The YLF/BTFC gene clusters have been reported as a marker of isolate geographic localisation [10,28]. Our data demonstrate a regional difference with BTFC cluster predominantly

found in the North West Region of Mount Isa and Mornington Island. This distribution is similar to previously reported Australian data [10]. Whereas, the remaining regions in our study are more evenly distributed, and therefore do not clearly demonstrate geographic localisation.

A previous Australian study from Darwin reported that patients with an *fhaB*3 positive isolate were twice as likely to be bacteraemic but not more likely to die [10]. Our data do not demonstrate an association with bacteraemia. It is possible that this discrepancy may be due to a lower number of cases in our cohort. However, the rate of bacteraemia in Darwin (56%) is lower than in this cohort (71%). Therefore, factors yet to be identified and not the *fhaB*3 gene are likely associated with bacteraemia in Queensland.

The proportion of LPS A and B genotypes in Queensland is lower than previously reported in northern Australia [12]. However, similar to previous reports, there was no association with genotype and mortality. We also report a significant proportion of isolates which were unable to be genotyped. This finding warrants additional phenotypic and genotypic investigation.

The prevalence of phenotypic and genotypic antimicrobial resistance is this cohort was low. This is an expected finding as *B. pseudomallei* resistance to first line agents is reported to be rare [29].

There are several limitations in this study. Patient data were limited and therefore analysis of patient comorbidities and outcomes could not be performed. Phenotypic antimicrobial susceptibility data was not available for all isolates. Patient treatment and prior antibiotic exposure data were also not available. Additionally, while Townsville laboratory is the state reference laboratory, some isolates may not have been referred. Finally, regional differences may not be accurately represented due to the low number of cases in some areas.

## Conclusion

There are both clinical and bacterial genomic differences across Queensland, Australia. Furthermore, we reveal that while the case-fatality rate has not significantly improved, this may be in part due to an increasing proportion of cases in elderly patients. A strategic infection prevention strategy and increased awareness of melioidosis in this at-risk cohort may result in earlier diagnosis and improved outcomes.

## Supporting information

**S1 Table. Reference strains used to create custom virulence factor database.**
(DOCX)

**S2 Table. Metadata of the previously published genomes.**
(DOCX)

**S3 Table. A. Bivariate and multivariate analysis of factors associated with bacteraemia.** B. Bivariate and multivariate analysis of factors associated with mortality.
(DOCX)

**S4 Table. Incidence of melioidosis by Health Service.**
(DOCX)

**S5 Table. Comparison of demographic, clinical, and virulence factors in relation to bacteraemia.**
(DOCX)

**S6 Table. Bivariate associations with LPSA.**
(DOCX)

**S7 Table. Bivariate associations with fhaB3.**
(DOCX)

**S8 Table. Bivariate associations with YLF/BTFC.**
(DOCX)

**S9 Table. Bivariate associations with bimA$_{Bm}$.**
(DOCX)

**S1 Fig. Hospital and health services where the majority of melioidosis patients were identified (created using R with the hospital and health service area boundaries across Queensland as the base map https://qldspatial.information.qld.gov.au/catalogue/custom/detail.page?fid={A4661F6D-0013-46EE-A446-A45F01A64D46} [18]).**
(TIFF)

## Author Contributions

**Conceptualization:** Ian Gassiep, Robert Norton, Patrick N. A. Harris.

**Data curation:** Ian Gassiep, Michelle J. Bauer, Thom Cuddihy.

**Formal analysis:** Ian Gassiep, Delaney Burnard, Budi Permana, Michelle J. Bauer, Thom Cuddihy, Brian M. Forde, Mark D. Chatfield, Weiping Ling.

**Funding acquisition:** Ian Gassiep.

**Methodology:** Delaney Burnard.

**Resources:** Patrick N. A. Harris.

**Software:** Thom Cuddihy, Brian M. Forde, Weiping Ling.

**Supervision:** Mark D. Chatfield, Robert Norton, Patrick N. A. Harris.

**Writing – original draft:** Ian Gassiep, Delaney Burnard, Budi Permana, Michelle J. Bauer, Thom Cuddihy, Brian M. Forde, Mark D. Chatfield, Weiping Ling, Robert Norton, Patrick N. A. Harris.

**Writing – review & editing:** Ian Gassiep, Budi Permana, Michelle J. Bauer, Brian M. Forde, Mark D. Chatfield, Weiping Ling, Robert Norton, Patrick N. A. Harris.

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
