## [Decision Letter · Decision Letter 0]

5 Aug 2023

Dear Dr Gassiep,

Thank you very much for submitting your manuscript "Melioidosis Queensland: An analysis of clinical outcomes and genomic factors" for consideration at PLOS Neglected Tropical Diseases. As with all papers reviewed by the journal, your manuscript was reviewed by members of the editorial board and by several independent reviewers. In light of the reviews (below this email), we would like to invite the resubmission of a significantly-revised version that takes into account the reviewers' comments. 

We cannot make any decision about publication until we have seen the revised manuscript and your response to the reviewers' comments. Your revised manuscript is also likely to be sent to reviewers for further evaluation.

Sincerely,

Joseph M. Vinetz

Section Editor

Joseph Vinetz

Section Editor

Reviewer's Responses to Questions

**Key Review Criteria Required for Acceptance?**

**Methods**

-Are the objectives of the study clearly articulated with a clear testable hypothesis stated?

-Is the study design appropriate to address the stated objectives?

-Is the population clearly described and appropriate for the hypothesis being tested?

-Is the sample size sufficient to ensure adequate power to address the hypothesis being tested?

-Were correct statistical analysis used to support conclusions?

-Are there concerns about ethical or regulatory requirements being met?

Reviewer #1: The methods were clear. 

It was not clear what proportion of eligible cases were included ie. had available isolates for sequencing.

Reviewer #2: The study population and area needs more description

**Results**

-Does the analysis presented match the analysis plan?

-Are the results clearly and completely presented?

-Are the figures (Tables, Images) of sufficient quality for clarity?

Reviewer #1: Table 2: Were first nations individuals over-represented with respect to their proportion of the local population?

Line 197: Was there any temporospatial clustering of ST identified? Similarly, how closely are the included ST related in the context of broader B. pseudomallei phylogeny. 

Line 205: Was there any co-association of genomic virulence factors? Was there any temporospatial clustering?

214: BTFC in First nations – was this related to spatial distribution. Were FN populations over-represented with respect to the populations in the regions in which BTFC isolates were recovered. 

223: Did the single CAZ resistant isolate have a Hx of prior exposure? Similarly for the single SXT resistant isolate?

Line 239 – Please provide interpretation for the MICs

Reviewer #2: results can be presented in more detail using phylogenetic trees

**Conclusions**

-Are the conclusions supported by the data presented?

-Are the limitations of analysis clearly described?

-Do the authors discuss how these data can be helpful to advance our understanding of the topic under study?

-Is public health relevance addressed?

Reviewer #1: Line 256; What were the incidence rates for TSV vs. NW (too broad range in the figure)

Line 268; Has there been a decline in melioid RFs among local FN populations? Is that what this line is suggesting?

Reviewer #2: (No Response)

**Editorial and Data Presentation Modifications?**

Reviewer #1: Figure 1. How were the bands determined? Given the wide range in incidence rates it might be clearer to list them. 

Figure 1. Is there incidence data available for Cairns and Hinterland, & Torres and Cape to help provide context?

Reviewer #2: (No Response)

**Summary and General Comments**

Reviewer #1: This is an interesting and well written manuscript.

My main comment would be around the need to provide some broader context regarding the genomic diversity and how this fits into what is already known about the genomic diversity in Australia and internationally. Similarly ST typing of a highly genetically diverse organism like B. pseudomallei may not capture the relatedness of isolates. It would have been interesting to see a phylogeny with isolates from this study placed into a broader national/international context.

Reviewer #2: (No Response)

PLOS authors have the option to publish the peer review history of their article (what does this mean?). If published, this will include your full peer review and any attached files.

Reviewer #1: No

Reviewer #2: Yes: Chaitanya Tellapragada
---

## [Editor Report · Decision Letter 1]

3 Oct 2023

Dear Dr Gassiep,

We are pleased to inform you that your manuscript 'Melioidosis Queensland: An analysis of clinical outcomes and genomic factors' has been provisionally accepted for publication in PLOS Neglected Tropical Diseases.

Best regards,

Joseph M. Vinetz

Section Editor

Joseph Vinetz

Section Editor

---

## [Editor Report · Acceptance letter]

9 Oct 2023

Dear Dr Gassiep,

We are delighted to inform you that your manuscript, "Melioidosis Queensland: An analysis of clinical outcomes and genomic factors," has been formally accepted for publication in PLOS Neglected Tropical Diseases.

Best regards,

Shaden Kamhawi

co-Editor-in-Chief

Paul Brindley

co-Editor-in-Chief
